# A comparative study of RF heating of deep brain stimulation devices in vertical vs. horizontal MRI systems

**Jasmine Vu**[1,2☯], **Bhumi Bhusal**[2☯], **Bach T. Nguyen**[2], **Pia Sanpitak**[1,2], **Elizabeth Nowac**[3], **Julie Pilitsis**[4], **Joshua Rosenow**[5], **Laleh Golestanirad**[1,2]*

1 Department of Biomedical Engineering, McCormick School of Engineering, Northwestern University, Evanston, Illinois, United States of America, 2 Department of Radiology, Feinberg School of Medicine, Northwestern University, Chicago, Illinois, United States of America, 3 Illinois Bone and Joint Institute (IBJI), Wilmette, Illinois, United States of America, 4 Department of Neurosciences & Experimental Therapeutics, Albany Medical College, Albany, New York, United States of America, 5 Department of Neurosurgery, Feinberg School of Medicine, Northwestern University, Chicago, Illinois, United States of America

☯ These authors contributed equally to this work.

* laleh.rad1@northwestern.edu

**Data Availability Statement:** All relevant data are within the paper and its Supporting Information files.

## Abstract

The majority of studies that assess magnetic resonance imaging (MRI) induced radiofrequency (RF) heating of the tissue when active electronic implants are present have been performed in horizontal, closed-bore MRI systems. Vertical, open-bore MRI systems have a 90˚ rotated magnet and a fundamentally different RF coil geometry, thus generating a substantially different RF field distribution inside the body. Little is known about the RF heating of elongated implants such as deep brain stimulation (DBS) devices in this class of scanners. Here, we conducted the first large-scale experimental study investigating whether RF heating was significantly different in a 1.2 T vertical field MRI scanner (Oasis, Fujifilm Healthcare) compared to a 1.5 T horizontal field MRI scanner (Aera, Siemens Healthineers). A commercial DBS device mimicking 30 realistic patient-derived lead trajectories extracted from postoperative computed tomography images of patients who underwent DBS surgery at our institution was implanted in a multi-material, anthropomorphic phantom. RF heating around the DBS lead was measured during four minutes of high-SAR RF exposure. Additionally, we performed electromagnetic simulations with leads of various internal structures to examine this effect on RF heating. When controlling for RMS $B_1^+$, the temperature increase around the DBS lead-tip was significantly lower in the vertical scanner compared to the horizontal scanner (0.33 ± 0.24˚C vs. 4.19 ± 2.29˚C). Electromagnetic simulations demonstrated up to a 17-fold reduction in the maximum of 0.1g-averaged SAR in the tissue surrounding the lead-tip in the vertical scanner compared to the horizontal scanner. Results were consistent across leads with straight and helical internal wires. Radiofrequency heating and power deposition around the DBS lead-tip were substantially lower in the 1.2 T vertical scanner compared to the 1.5 T horizontal scanner. Simulations with different lead structures suggest that the results may extend to leads from other manufacturers.

**Funding:** This work was supported by the National Institutes of Health [grant numbers R03EB029587, R01EB030324, and T32EB025766]. The funders had no role in study design, data collection and analysis, decision to publish, or preparation of the manuscript.

**Competing interests:** The authors have declared that no competing interests exist.

## Introduction

Magnetic resonance imaging (MRI) provides excellent soft tissue contrast and clear visualization of fine anatomical structures without exposing the patient to ionizing radiation. Unfortunately, MRI is not as readily accessible to patients with active implantable medical devices (AIMDs), such as those with deep brain stimulation (DBS) implants, due to safety risks associated with radiofrequency (RF) heating of the tissue surrounding the implant. This is troubling, as up to 75% of patients with DBS systems will require an MRI exam during the lifetime of the device [1].

Excessive local heating in the tissue surrounding tips of AIMD leads arises due to a phenomenon commonly known as the *antenna effect*, where the electric field of the MRI transmit coil couples with long conductive leads and amplifies the specific absorption rate (SAR) of RF energy in the tissue surrounding the lead's tip [2–7]. Through this, patients have sustained thermal injuries and permanent neurological damage [8,9] leading to strict guidelines developed by DBS manufacturers. For example, MR-conditional DBS systems from Abbott Medical limit MRI exams to those performed in horizontal, closed-bore scanners at a 1.5 T magnetic field strength and with an RMS B1+ < 1.1 µT [10]. The RMS $B_1^+$ is a patient-independent metric of RF exposure and is the root mean square value of $B_1^+$ averaged over a period of 10 seconds [11]. Complying with these restrictions has proven to be difficult as clinical protocols that are optimized to visualize DBS targets or those in routine cardiac and musculoskeletal imaging have RMS $B_1^+$ values that far exceed these limits (S1 Table in S1 File). Although the RMS $B_1^+$ can be reduced by adjusting sequence parameters such as increasing the repetition time or reducing the flip angle, such adjustments can compromise image quality, contrast, and the total acquisition time.

To date, the majority of studies that have assessed RF heating of AIMDs have been performed in horizontal, closed-bore MRI scanners. Vertical, open-bore systems have a 90˚ rotated magnet and a fundamentally different RF coil geometry which produces a notably different electromagnetic (EM) field distribution within the human body [12,13]. Little is known about the RF heating of AIMDs in this class of scanners which are now available at higher field strengths (e.g., 1.2 T) and capable of high resolution anatomical and functional imaging. Recently, our group performed simulation studies with simplified DBS lead models—most cases represented lead-only DBS systems—which showed that the local 0.1g-averaged SAR around the tips of wires following typical DBS lead trajectories was lower in a vertical scanner compared to a conventional, horizontal scanner [12,13]. The current work builds on our previous proof-of-concept studies to answer two major open questions. First, we set to determine whether simulation results of SAR around simplified wire models would translate to measured temperature rise around commercial DBS devices in an MRI environment. This is important because simulations do not account for the complexities of the internal geometry of realistic leads. Specifically, commercial AIMD leads have several interwoven helical micro wires which exhibit different electric lengths depending on the pitch of the helix and therefore, behave differently when exposed to the MRI electric fields [14]. To examine if previously reported simulation results will be confirmed experimentally, we measured the RF heating of a commercial Abbott Medical DBS device implanted in an anthropomorphic phantom following 30 new patient-derived configurations during MRI in a 1.2 T vertical scanner (OASIS, Fujifilm Healthcare, Tokyo, Japan) and compared it with RF heating generated in a 1.5 T conventional, horizontal scanner (Aera, Siemens Healthineers, Erlangen, Germany). The previous study only evaluated four trajectories with a DBS device from Medtronic [13]. Second, we explored whether the observed experimental results could potentially extend to leads from other manufactures, that is, leads with different electric lengths. To do this, we performed EM simulations

with lead models of various internal structures where the pitch of the helical internal wire was varied to generate different electrical lengths. We compared the power deposition in the tissue around the tips of the leads with different internal wire lengths during MRI in a 1.5 T horizontal scanner and a 1.2 T vertical scanner.

This work is the first large-scale experimental evaluation of RF heating of DBS devices during MRI in a 1.2 T vertical scanner compared to RF heating in a conventional 1.5 T scanner. Additionally, our theoretical groundwork on the effect of the lead's internal geometry provides the first evidence for the possibility of extrapolating the outcome to leads from other manufacturers.

## Materials and methods

### Creation of patient-derived DBS lead trajectories mimicking *in vivo* scenarios

It is established that RF heating of an elongated implant (such as leads in cardiovascular and neuromodulation devices) is largely affected by the implant's position within the human body and its orientation with respect to the MRI electric fields [6,15–19]. Therefore, studies that aim to assess RF heating of leads should ideally do so by replicating patient-derived device configurations in an environment that mimics the *in vivo* scenario. To do this, we identified clinically relevant lead trajectories from postoperative computed tomography (CT) images of 30 patients who underwent DBS surgery in our institutions from May 2017 to September 2020. The RF heating in a vertical MRI scanner of these 30 trajectories has not been previously studied. The retrospective use of patients' imaging data for the purpose of modeling and simulation was approved by Northwestern Memorial Hospital and Albany Medical Center's institutional review boards. DBS lead trajectories for this study can be found in S2 File.

Lead trajectories were segmented from CT images using 3D Slicer 4.10.2 (http://slicer.org) and processed in a CAD tool (Rhino 6.0, Robert McNeel & Associates, Seattle, WA) to create 3D-printed guides that helped to accurately position a commercial DBS device along different trajectories (Fig 1). Once the leads were positioned in place, the guides were removed from the phantom so that their presence did not affect the heating experiments.

To provide a more realistic replication of the electric field distribution around the implanted lead, we used a multi-material anthropomorphic phantom consisting of a body-shaped container and a refillable skull structure. The phantom design was based on CT images of a patient with a DBS device [20]. The skull was filled with a tissue mimicking gel ($\sigma$ = 0.40 S/m, $\varepsilon_r$ = 79, similar to values reported for brain tissue), [21] prepared by mixing 32 g/L of edible agar (Landor Trading Company, gel strength 900 g/cm$^2$) with saline solution (2.25 gNaCl/L). The remaining head-torso component of the phantom was filled with 18 L of saline solution ($\sigma$ = 0.50 S/m, $\varepsilon_r$ = 80) mimicking the conductivity of the average tissue. Using an agar-based solution to fill the skull was advantageous compared to using polyacrylamide gel as it formed a semi-solid gel which kept the leads in place. The thermal conductivity of the solidified agar gel was ~0.56 J/K-S [22] which was similar to that of grey matter [23].

To further assess the degree to which RF exposure of DBS devices implanted in the anthropomorphic phantom represented the *in vivo* scenario, we performed EM simulations to calculate the distribution of the MRI-induced electric fields on various coronal planes inside our phantom and compared them with the electric fields inside a heterogenous human body model consisting of 32 tissue classes from ANSYS (ANSYS, Canonsburg, PA) (S1 and S2 Figs in S1 File). Results demonstrated a good agreement between the electric field distributions in the phantom and the heterogeneous body model (S3 and S4 Figs in S1 File) ensuring that the experimental results in the anthropomorphic phantom are a reliable indicator of RF heating in

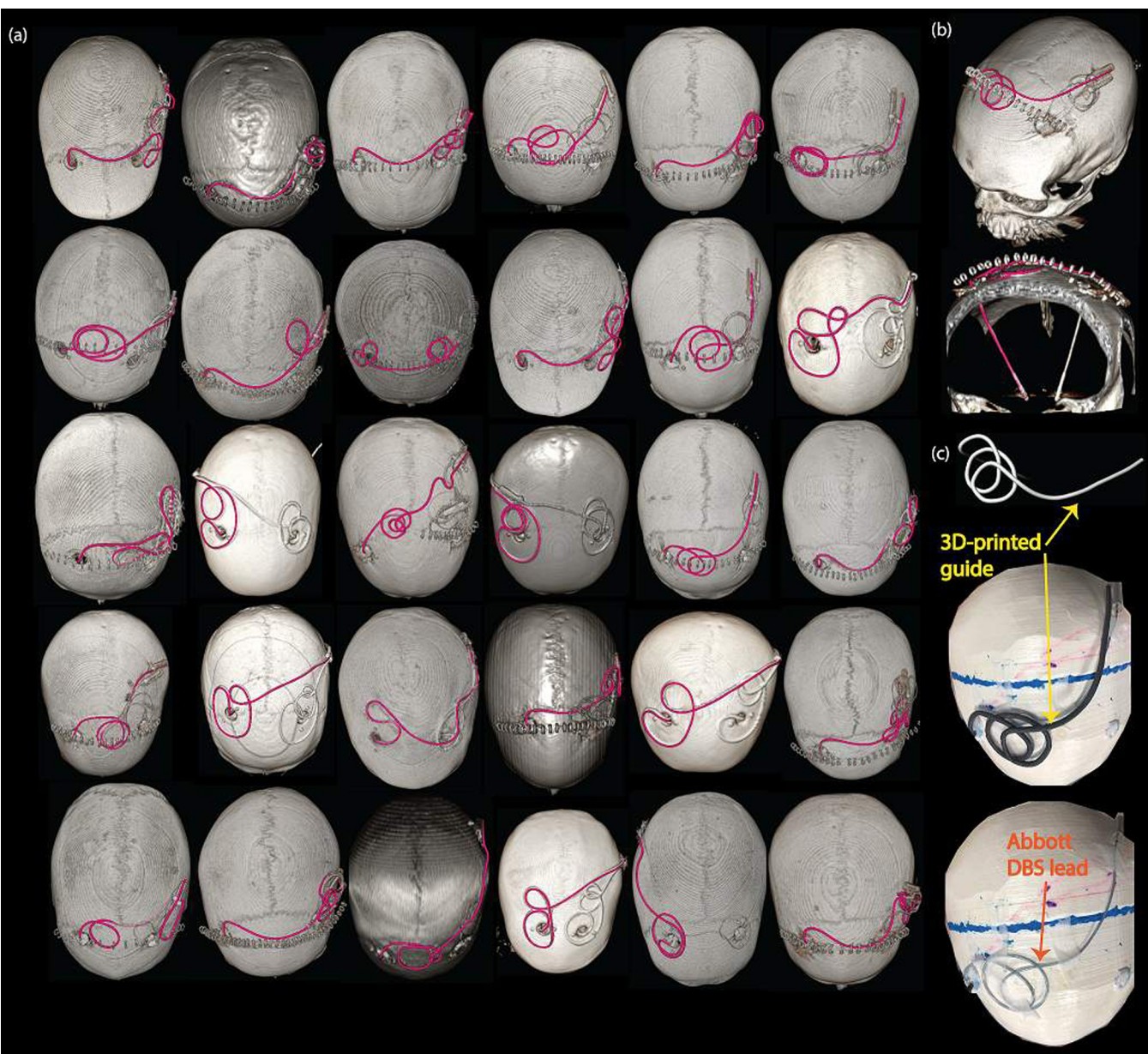

**Fig 1. Evaluated DBS lead trajectories.** (a) Thirty DBS lead trajectories were evaluated in this study. The trajectories are highlighted (magenta) on the 3D surface rendered views of computed tomography images of patients with implanted DBS leads. (b) Example segmentation of the DBS lead trajectory. (c) Example 3D printed model of a lead trajectory for replication in RF heating experiments with a commercial DBS system.

patients. This is important, as recent studies have highlighted that the electric fields (and by proxy, SAR and RF heating) inside the box-shaped ASTM phantom can significantly differ from fields that are induced in the human body [24].

## RF heating experiments

A full commercial DBS system from Abbott (Abbott Medical, Plano, TX) consisting of a 40 cm lead (model 6173), a 50 cm extension (model 6371), and an implantable pulse generator (IPG) (Infinity 6660) was implanted in the anthropomorphic phantom. The DBS lead was inserted

into the skull with fluoroptic temperature probes securely attached to the two most distal electrode contacts. Placement of the lead and temperature probes inside the skull emulated the location and angle of insertion for targeting the subthalamic nucleus (STN). The lead was connected to the extension and the IPG with the extension routed laterally along the neck and the IPG placed in the pectoral region. The IPG was turned off during the experiments.

Thirty unique, clinically relevant trajectories were replicated during experiments at the vertical and horizontal scanners. For all trajectories, the lead was implanted to target the right STN; 27 trajectories were contralateral to the IPG (i.e., IPG was placed in the left pectoral region) while 3 trajectories were ipsilateral to the IPG.

RF heating experiments were performed in a 1.5 T Aera horizontal scanner (Siemens Healthineers, Erlangen, Germany) and in a 1.2 T Oasis vertical scanner (Fujifilm Healthcare, Tokyo, Japan) (Fig 2) using the body transmit coils at both scanners. The phantom was placed in the head-first, supine position, and an imaging landmark at the level of the DBS lead-tip was selected for all experiments. RF exposure was generated using high-SAR turbo spin echo (TSE) and fast spin echo (FSE) sequences such that the RMS $B_1^+$ was 4 μT at both scanners (Table 1). Each experimental configuration included only one DBS lead with a single lead trajectory, representing cases of unilateral DBS. Temperature rise during RF exposure was measured at the DBS lead-tip using temperature probes (OSENSA, BC, Canada). The temperature was recorded continuously throughout the RF exposure for the total acquisition time (TA) of 224 seconds. The maximum temperature rise ($\Delta T_{max}$) was quantified as the difference between the baseline temperature at the onset of RF exposure and the highest measured temperature. The setup was allotted ample time to return to the baseline temperature prior to evaluating the next lead trajectory.

## Investigating the effect of the lead's length and internal structure

DBS leads from different manufacturers consist of internal helical wires that are wound at different pitches and thus, have different electrical lengths even when the apparent length of the lead seems to be the same for different lead models. This is important to consider because the RF heating of an elongated implant is a resonance phenomenon which depends on the length of the lead [25,26]. Therefore, we performed EM simulations with leads of three different internal structures—straight and helical wires—to assess whether the results of our experiments could potentially extend to other models of DBS devices (Fig 3). This allowed us to examine if the difference in RF heating was specific to the electrical length of the lead used in our experiments or if it was a trend that could be observed for leads of shorter and/or longer lengths.

## Simulation setup

Electromagnetic simulations were implemented in ANSYS Electronic Desktop 2021 R1 HFSS using three lead trajectories that generated a small, median, and large difference in $\Delta T_{max}$ between RF exposures in the horizontal versus vertical scanners (Fig 3). The coordinates along the lead trajectory were extracted during image segmentation and were used to reconstruct the model of the DBS lead. The lead and extension consisted of a core wire made of platinum-iridium ($\sigma = 4 \times 10^6$ S/m) embedded within a urethane insulation ($\sigma = 0$ S/m, $\varepsilon_r = 3.5$). The apparent length of the modeled lead and extension was 90 cm to match the commercial device used during experiments; however, the electrical length of the core wire was changed by modeling either a straight wire or helical wires with pitches of 1 and 2 mm. The full DBS system was implanted in a standard homogeneous model of the human body truncated at the abdomen ($\sigma = 0.40$ S/m, $\varepsilon_r = 79$) where a triangulated surface model of the patient's head and the DBS system were manually aligned to the standard body model via rigid transformation/registration (6 degrees of freedom) to place the device in the correct anatomical position.

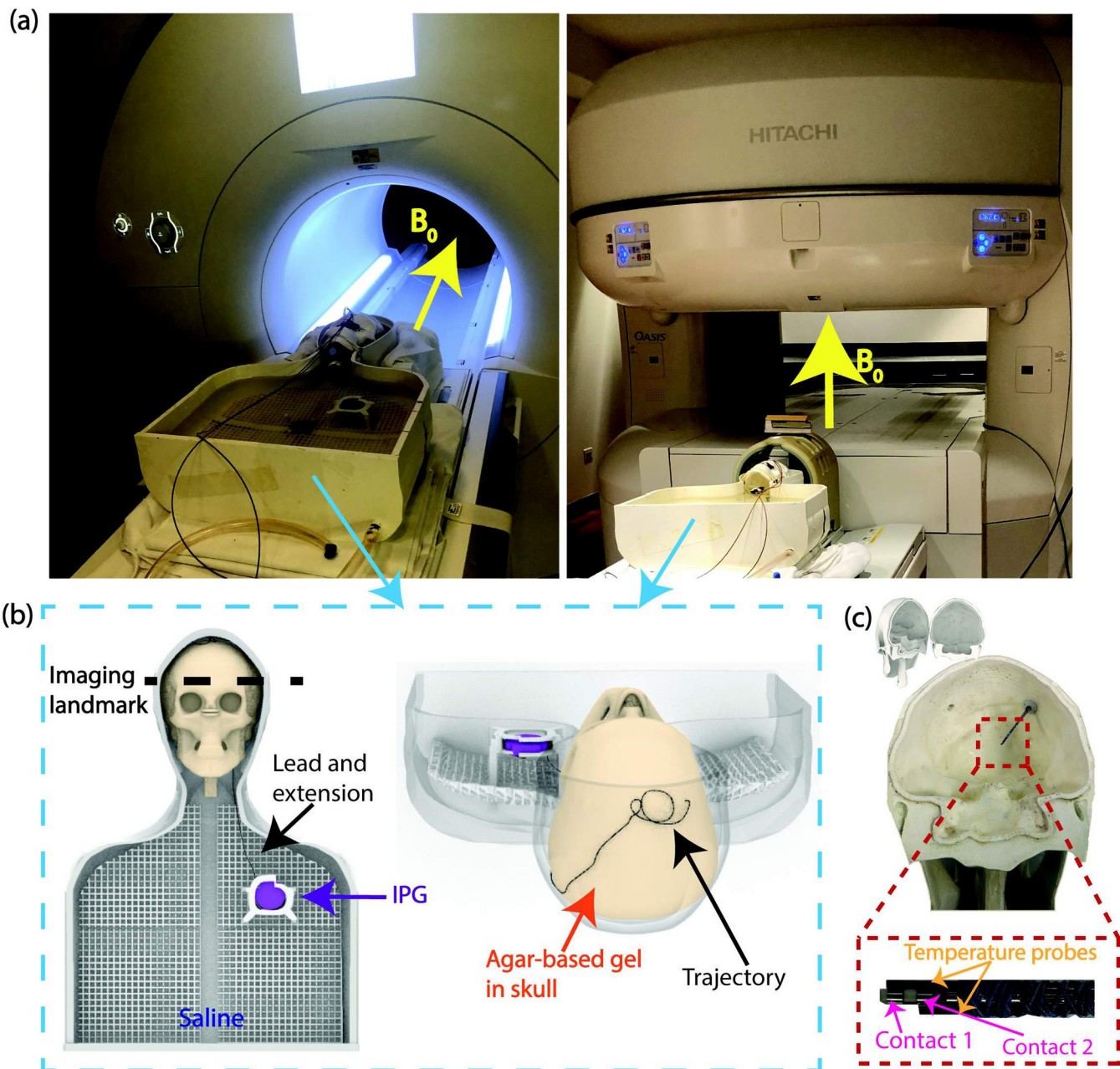

**Fig 2. Experimental setup.** (a). Anthropomorphic phantom with full DBS system implanted in the 1.5 T Aera horizontal scanner (left) and in the 1.2 T Oasis vertical scanner (right). (b) Rendering of anthropomorphic phantom components and a full DBS system implanted in the phantom. (c) 3D-printed skull with the DBS lead and temperature probes inserted. Temperature probes were attached to the lead to measure temperature rise at the distal end.

A high-pass 16-rung horizontal birdcage coil tuned to 63.6 MHz was constructed based on the details provided by Siemens (Fig 4A). Similarly, a numerical model of a radial planar 12-rung birdcage coil with the specifications of the body coil in the Oasis scanner tuned to 50.4 MHz was constructed based on the data provided by the manufacturer (Fig 4B). Both coils provided quadrature excitation with ports separated by 90°; the vertical coil had four ports while the horizontal coil had two ports. The input voltage applied to each port was adjusted to generate a mean $B_1^+$ of 4 µT on a central transverse plane passing through the center of the coil.

**Table 1. MRI pulse sequences.**

| Sequence Parameter | T2W TSE at 1.5 T Aera | T2W FSE at 1.2 T Oasis |
|---|---|---|
| TE (ms) | 96 | 96 |
| TR (ms) | 2780 | 2728 |
| Matrix size | 512 x 512 | 512 x 512 |
| Acquisition Time (sec) | 224 | 224 |
| RMS $B_1^+$ (µT) | 4 | 4 |

We also compared the orientation of the incident electric field with respect to the lead trajectory between the horizontal and vertical coils. Electromagnetic simulations were performed with the human body model without any implant (mean $B_1^+$ of 4 µT) to calculate the tangential component of the incident electric field ($E_{tan}$) along the selected DBS lead trajectory at different time points since the orientation of the electric field changes as the field rotates (Eq 1).

$$E_{\text{tan}}(\bar{r}, t) = \overrightarrow{E}(\bar{r}, t) \cdot \hat{a}(\bar{r}) \tag{1}$$

$\vec{E}$ represents the incident electric field, and $\hat{a}$ is the unit vector tangential to the DBS lead trajectory. Secondly, we calculated the peak-to-peak value of the induced voltage along the first 10 cm of the extracranial portion of the DBS lead trajectory ($V_{pp}$) (Eq 2).

$$V_{pp} = \int_P^Q E_{tan}(\bar{r}, t) d\bar{r} \Big|_{\text{peak-to-peak}} \tag{2}$$

Power deposition in the tissue adjacent to the DBS lead-tip was quantified using the SAR calculation module incorporated in HFSS. The maximum of the 0.1g-averaged SAR, 0.1gSAR-max, was calculated in a $(20 \text{ mm})^3$ cubic tissue region surrounding the lead-tip. A fine mesh resolution was enforced within this volume, where the maximum tetrahedral mesh edge length was 2 mm for the tissue region around the DBS lead-tip and 0.5 mm for the DBS lead. Numerical convergence was ensured by imposing a constraint on the maximum variation of the scattering parameters between two consecutive iterations [27].

## Statistical analysis

A one-tailed Wilcoxon signed rank test was conducted to assess the difference in the measured $\Delta T_{max}$ between RF exposure experiments in the 1.2 T vertical and 1.5 T horizontal scanners. Statistical significance was established for $p < 0.05$. Statistical analysis was performed in MATLAB 2020b (The MathWorks Inc., Natick, MA).

## Demographics of patients

In total, this study included DBS lead trajectories from 30 patients (21 men) with a mean age ± standard deviation of 60.4 ± 13.5 years (Table 2). The most common DBS indication and target were Parkinson's disease (PD) and the STN, respectively.

## Results

### Peak-to-peak induced voltage

$V_{pp}$ was calculated for the lead trajectory that generated a large difference in $\Delta T_{max}$ between RF exposures in the horizontal versus vertical scanners. The $V_{pp}$ for the first 10 cm of the extracranial portion of this lead trajectory in the horizontal coil was 3.5 V and 1.3 V in the vertical

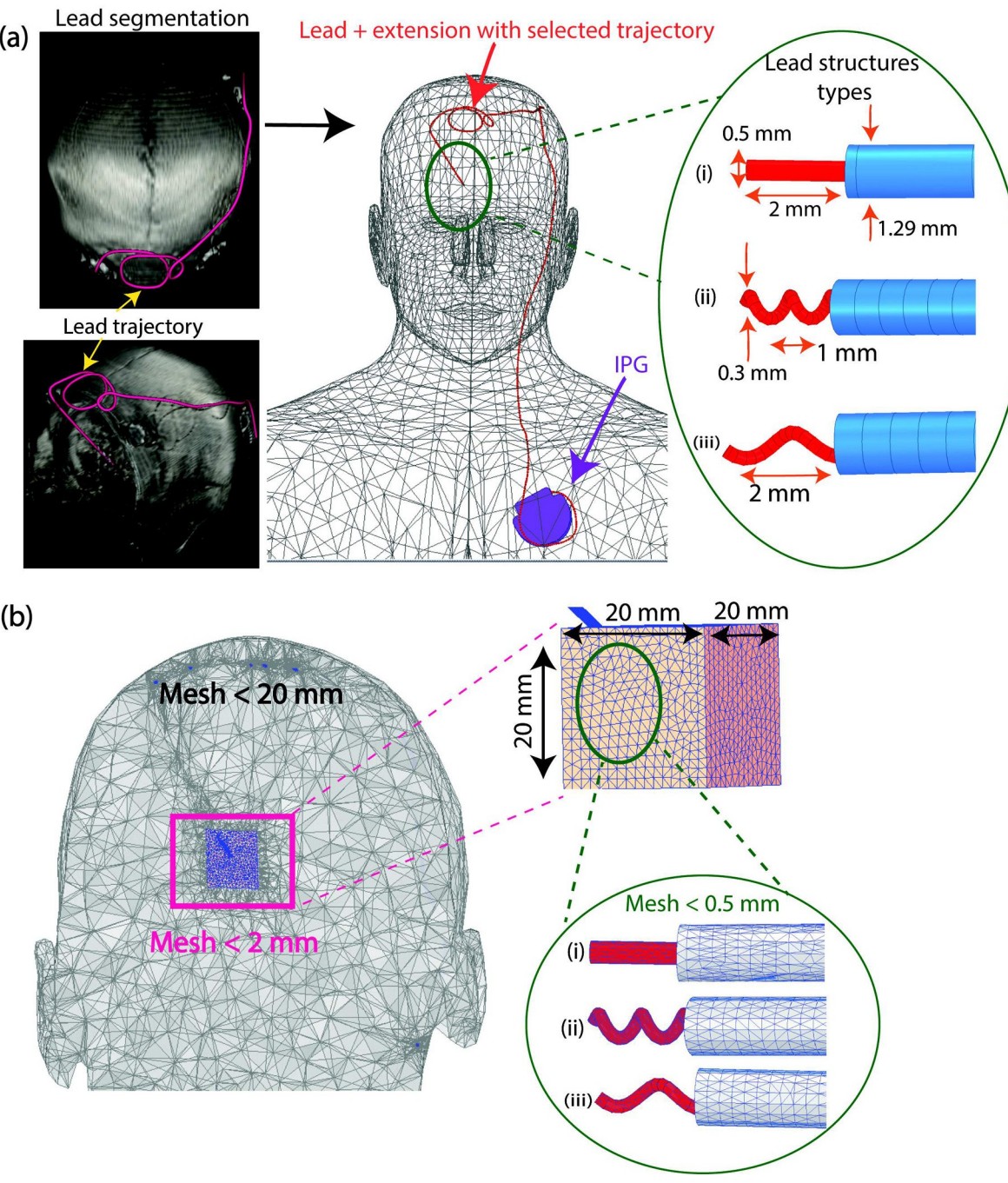

**Fig 3. DBS system modeling and simulation setup.** (a) Segmentation of an example DBS lead trajectory (magenta) from the lead artifact displayed in the 3D surface rendered view of a patient's computed tomography image and reconstruction of a full DBS model oriented in a homogeneous body model truncated at the abdomen. Reconstructed DBS lead models of the same trajectory (Pt-Ir core wire (red) within a urethane insulation (blue)) with various internal geometries were evaluated: (i) straight wire, (ii) helical wire with a 1 mm pitch, and (iii) helical wire with a 2 mm pitch. (b) Example mesh distributions of the body, tissue region for specific absorption rate (SAR) calculations, and the lead.

coil, demonstrating lower $V_{pp}$ with lower RF heating. Fig 4C illustrates the $E_{tan}$ values and the incident electric field along the entire lead trajectory in both coils at time t = 0. Additionally,

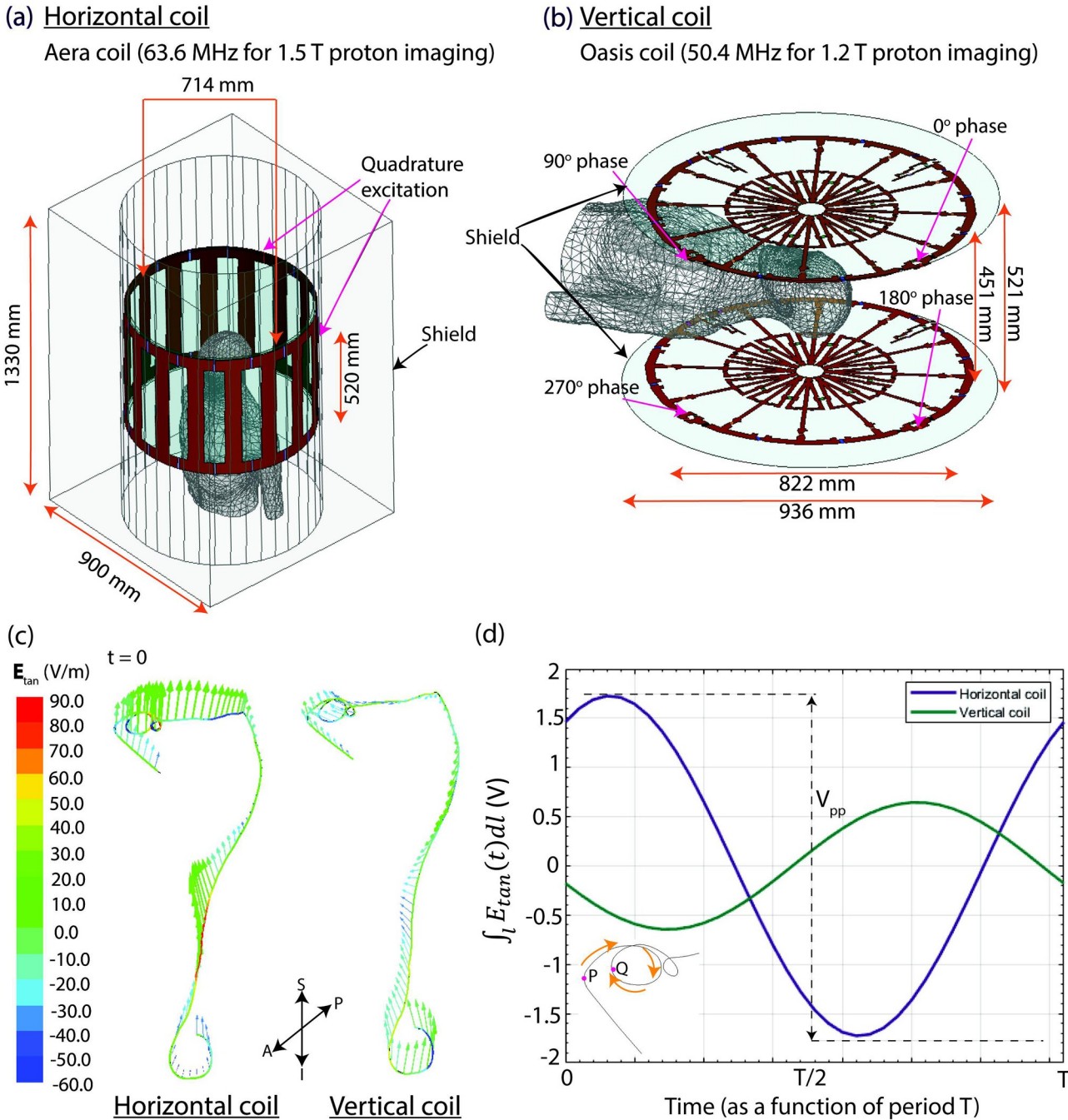

**Fig 4. RF coil models and $E_{tan}$ distribution.** (a) Numerical models of the 1.5 T Aera horizontal birdcage coil and (b) the 1.2 T Oasis radial planar birdcage coil. (c) The colormaps show the distribution of $E_{tan}$ and the green arrows indicate the incident electric field along the entire lead trajectory in the horizontal and vertical coils at time t = 0. (d) Time evolution of the induced voltage in the horizontal and vertical coils for the lead trajectory between points P and Q. Point P is at the location where the DBS lead exits the skull, and Point Q is 10 cm further along the extracranial portion of the lead.

Fig 4D shows the induced voltage of the first 10 cm of the extracranial portion of the lead trajectory in each coil through the time cycle.

**Table 2. Summary of patient population.**

| Parameter | Value[a] |
|---|---|
| Number of patients | 30 |
| Mean Age (y) | 60.4 ± 13.5 (21–76) |
| Women | 57. 0 ± 22.0 (21–76) |
| Men | 61.9 ± 7.8 (43–71) |
| Sex | |
| Women | 9 |
| Men | 21 |
| DBS Indication | |
| Parkinson's disease | 21 |
| Parkinson's disease and dystonia | 2 |
| Dystonia | 1 |
| Cervical dystonia | 1 |
| Essential tremor | 1 |
| Essential tremor and Parkinson's disease | 1 |
| Orthostatic tremor | 1 |
| Obsessive compulsive disorder | 2 |
| DBS target | |
| Subthalamic nucleus | 21 |
| Globus pallidus internus | 5 |
| Ventral intermediate nucleus of the thalamus | 2 |
| Ventral capsule/ventral striatum | 1 |
| Anterior limb of the internal capsule | 1 |

[a]Unless otherwise specified, data are number of participants.

## Experimental temperature measurements

Across all the trajectories, $\Delta T_{max}$ was significantly lower for RF exposure in the vertical scanner compared to RF exposure in the conventional horizontal scanner when the input power of each scanner was adjusted to generate the RMS $B_1^+$ of 4 μT (p = $9.13 \times 10^{-7}$). In the vertical scanner, the range of $\Delta T_{max}$ was 0.04–1.01˚C with a mean ± standard deviation of 0.33 ± 0.24˚C while the range of $\Delta T_{max}$ was 1.84–12.92˚C with a mean ± standard deviation of 4.19 ± 2.29˚C in the horizontal scanner. Fig 5 illustrates the temperature profiles for the different lead trajectories throughout the duration of the MR sequences in the two scanners, violin plots of the $\Delta T_{max}$ distributions, and $\Delta T_{max}$ for each trajectory.

## Simulated power deposition in the tissue

Electromagnetic simulations were performed with three different DBS lead trajectories, identified as those that demonstrated a small, median, and large difference between $\Delta T_{max}$ measured in the vertical and horizontal scanners. A total of nine pairs of simulations were performed. For the selected lead trajectories, we evaluated the effect of the lead's internal geometry on the power deposition in the tissue surrounding the DBS lead-tip. The electrical lengths of the inner conductors were 90 cm for the straight wire, 116 cm for the helical wire with a pitch of 2 mm, and 171 cm for the helical wire with a pitch of 1 mm. We observed the same trend of reduced RF heating in the vertical scanner for all cases. For a mean $B_1^+$ = 4 μT generated over an axial plane at the center of the imaging region, the range of $0.1gSAR_{max}$ was 82.6–274.6 W/kg in the vertical coil compared to 1027.5–2949.9 W/kg in the horizontal coil. The

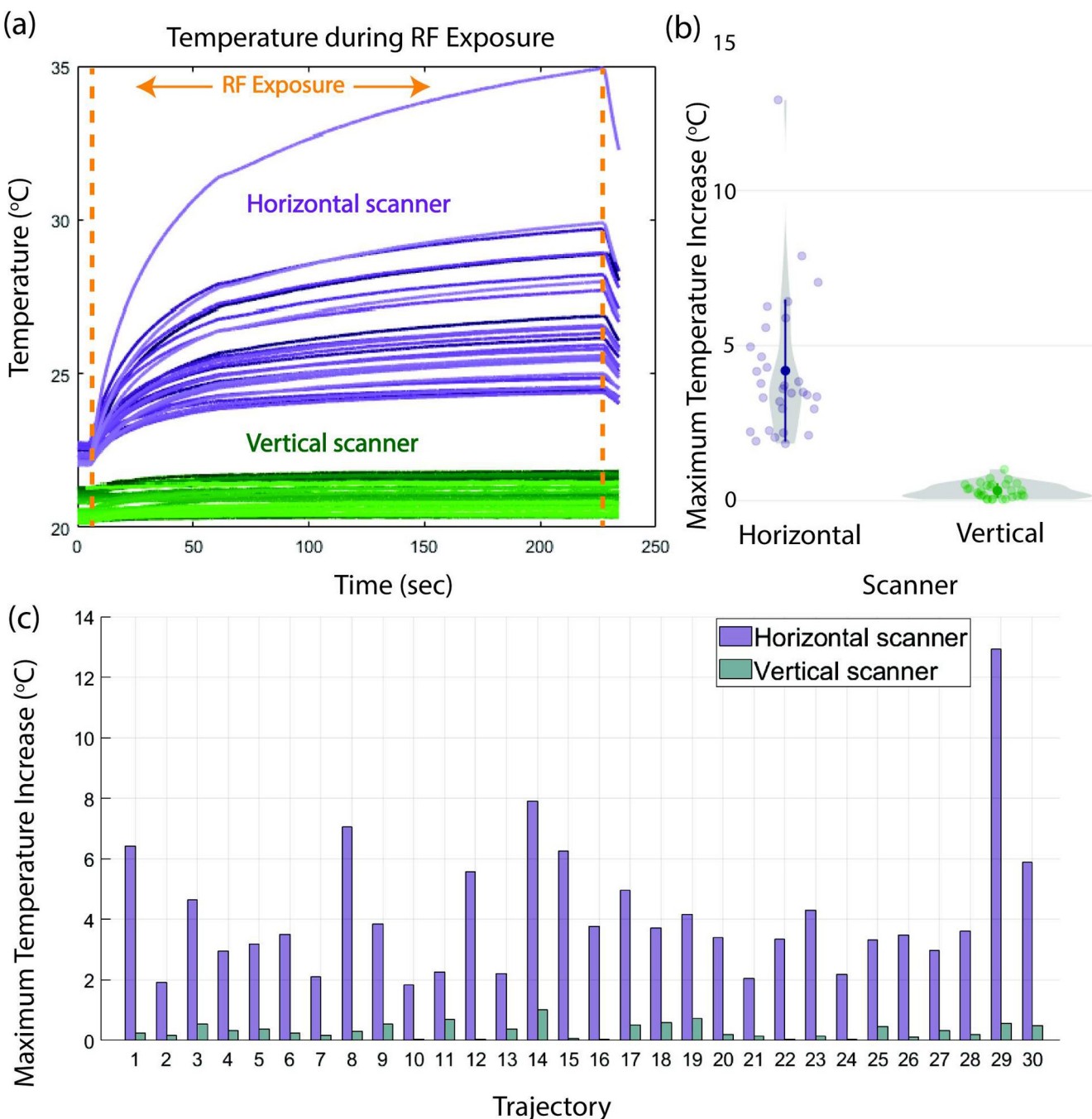

**Fig 5. Experimental temperature results.** (a) Measured temperature at the DBS lead-tip before, during, and after radiofrequency (RF) exposure in the horizontal and vertical scanners. (b) Violin plots of the maximum temperature increase in the horizontal and vertical scanners. The shaded circles and sprouting lines represent the mean and ± one standard deviation, respectively. (c) The maximum temperature increase during RF heating experiments for each trajectory.

mean ± standard deviation of $0.1gSAR_{max}$ was $166.7 \pm 63.8$ W/kg and $2216.7 \pm 693.7$ W/kg for the vertical and horizontal coils, respectively (Fig 6).

## Discussion

Current MRI guidelines for internalized DBS systems require the use of horizontal, closed-bore scanners, limiting potential applications of MRI in a continuously expanding population of patients with DBS. The safety risk of localized RF-induced tissue heating is a well-known barrier to MRI for patients with DBS implants. Recently, increased engineering efforts have targeted this problem through alteration of the design and methodologies of DBS and MRI. For example, MRI hardware modification has been proposed to reduce the antenna effect by shaping the electric field of the scanner through parallel transmit [19,28–31] or reconfigurable MRI technology [32–35]. Changes in DBS lead material [36,37] and modification of surgical lead implantation [6] have also been explored as alternative approaches. Although promising in theory, none of these techniques have found their way to the clinic yet, and there are ongoing efforts to enable implant-friendly MRI.

RF heating of elongated implants (such as leads in electronic devices) in an MRI environment is the result of coupling of the transmit electric field of the MRI RF coil with conductive wires of the implant. The efficiency of this coupling is directly affected by the magnitude of the electric field [38] as well as the orientation of the E-field vector with respect to the wire [18,25]. We calculated the peak-to-peak value of the induced voltage as a metric to compare the RF heating in the different scanner types. Since vertical MRI scanners have a 90˚ rotated RF coil, the E-field induced in the human body is substantially different from that of conventional birdcage coils. The distributions of Etan along the lead trajectory and the calculated $V_{pp}$ in the two coils (Fig 4C and 4D) also illustrate this difference with a 10-fold reduction in the $V_{pp}$ in the vertical coil. Further, Kazemivalipour et al. simulated the maximum SAR around the DBS lead-tip in a horizontal birdcage coil tuned to 50.4 MHz to match the Larmor frequency of the Oasis vertical scanner [13]. This comparison showed that the maximum SAR around the DBS lead-tip was still greater in the horizontal coil tuned to 50.4 MHz compared to the vertical coil tuned to 50.4 MHz; the reduction in SAR is mostly due to the different E-field orientations rather than the difference in resonance frequencies.

Recent simulation studies showed that vertical scanners could generate lower SAR around DBS lead models compared to conventional scanners; however, these results have not been rigorously examined in experiments with patient-derived lead trajectories. In this present study, we measured the RF heating of a commercial DBS system implanted in an anthropomorphic phantom following 30 unique patient-derived lead trajectories in the 1.2 T Oasis vertical scanner compared to the 1.5 T Aera horizontal scanner. The average measured $\Delta T_{max}$ was reduced by 12-folds at the vertical scanner compared to the horizontal scanner. For a high SAR sequence (RMS $B_1^+$ = 4 μT), temperature increase was well below 2˚C for all trajectories in the vertical scanner whereas $\Delta T_{max}$ up to 12˚C was recorded in the horizontal scanner. Heating in both scanners was noticeably different across trajectory-related parameters; future work is needed to determine which trajectory-related parameters (i.e. number, size, and position of extracranial loops) contribute to the difference in heating between the two scanners.

Additionally, we performed numerical simulations with leads of different internal geometries—and hence different electrical lengths—to investigate whether the results of our experiments could potentially extend to leads from other manufacturers. This is important because the internal wires of most DBS leads have a helical structure, both to increase mechanical flexibility and as a strategy to increase the electric inductance which can ultimately reduce MR-induced RF currents [39,40]. This means that the electrical length of internal wires is usually different from the apparent length of the lead (i.e., internal wires of a 40-cm DBS lead are much longer than 40 cm). For this reason, leads from different manufacturers—or even different lead models from the same manufacturer—do not necessarily behave similarly in an MRI

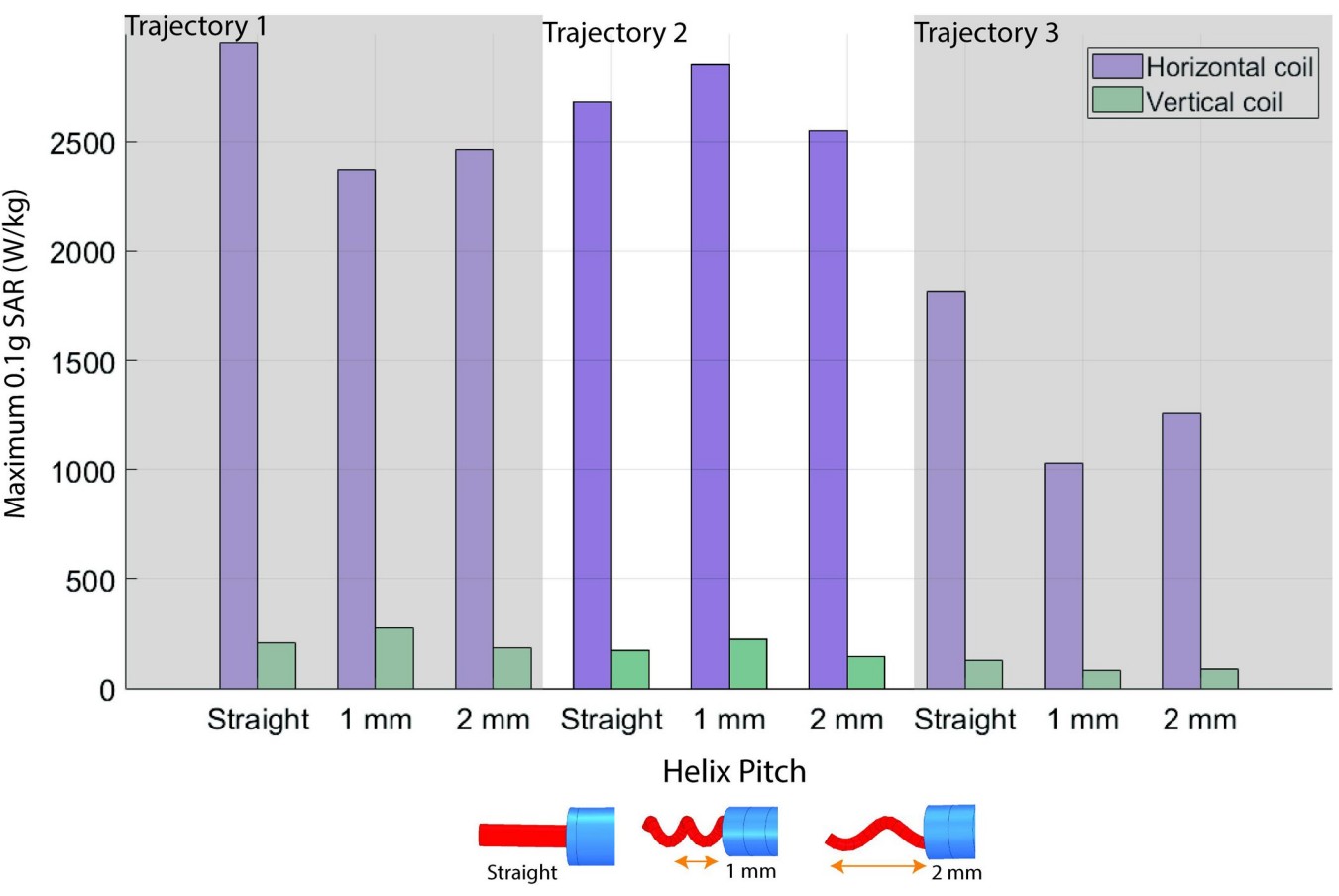

**Fig 6. Simulated power deposition of leads with different electrical lengths.** Simulated maximum of the 0.1g-averaged SAR at the DBS lead-tip for leads with different internal wire geometries across three different lead trajectories.

environment. Our simulations modeled leads with electrical lengths of 90–171 cm, demonstrating that the vertical scanner generated a substantially lower SAR at the lead's tip compared to the horizonal scanner in all these cases regardless of the lead trajectory. These simulation results suggest that our experimental results here could potentially generalize to DBS devices from other manufacturers.

## Conclusion

In conclusion, we demonstrate that RF exposure from a vertical MR scanner induces significantly less heating on DBS devices than a conventional, horizontal scanner. Our experimental results show that measured temperature increase did not exceed 2°C in the vertical scanner even when a high-power sequence was applied. Similarly, simulation results suggest that the benefits of vertical MRI for reducing RF heating may apply to other DBS lead models than the one used in this set of experiments.

## Supporting information

**S1 File. Supplementary material to the manuscript.**
(PDF)

**S2 File. DBS lead trajectories.**
(AEDTZ)

## Author Contributions

**Conceptualization:** Jasmine Vu, Bhumi Bhusal, Laleh Golestanirad.

**Data curation:** Jasmine Vu, Bhumi Bhusal, Laleh Golestanirad.

**Formal analysis:** Jasmine Vu, Bhumi Bhusal, Laleh Golestanirad.

**Funding acquisition:** Laleh Golestanirad.

**Investigation:** Bhumi Bhusal, Bach T. Nguyen, Pia Sanpitak, Laleh Golestanirad.

**Methodology:** Jasmine Vu, Bhumi Bhusal, Bach T. Nguyen, Pia Sanpitak, Laleh Golestanirad.

**Resources:** Elizabeth Nowac, Julie Pilitsis, Joshua Rosenow, Laleh Golestanirad.

**Supervision:** Laleh Golestanirad.

**Writing – original draft:** Jasmine Vu, Bhumi Bhusal, Laleh Golestanirad.

**Writing – review & editing:** Jasmine Vu, Bhumi Bhusal, Laleh Golestanirad.

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
