## [Decision Letter · Decision Letter 0]

14 Sep 2022

PONE-D-22-22994A comparative study of RF heating of deep brain stimulation devices in vertical vs. horizontal MRI systems

PLOS ONE

Dear Dr. Golestanirad,

Thank you for submitting your manuscript to PLOS ONE. After careful consideration, we feel that it has merit but does not fully meet PLOS ONE’s publication criteria as it currently stands. Therefore, we invite you to submit a revised version of the manuscript that addresses the points raised during the review process.

The manuscript was reviewed by two experts in the field. While both experts highly value the potential impact of the work and call the manuscript well written, they both remark in full agreement that the number of trajectories for the vertical MRI system is too low to extend the conclusion of the paper. Furthermore, the reviewers see issues with normalization and reporting of results (for example temperature curves, induced EMF). Nevertheless both reviewers are very positive about the work and encourage a thorough review.

We look forward to receiving your revised manuscript.

Kind regards,

Stephan Orzada

Academic Editor

PLOS ONE

“This work was supported by the National Institutes of Health [grant numbers R03EB029587, R01EB030324, and T32EB025766].”

Please include your amended statements within your cover letter; we will change the online submission form on your behalf."

Reviewers' comments:

Reviewer's Responses to Questions

**Comments to the Author**

1. Is the manuscript technically sound, and do the data support the conclusions?

Reviewer #1: Partly

Reviewer #2: Yes

2. Has the statistical analysis been performed appropriately and rigorously? 

Reviewer #1: N/A

Reviewer #2: Yes

3. Have the authors made all data underlying the findings in their manuscript fully available?

Reviewer #1: Yes

Reviewer #2: No

4. Is the manuscript presented in an intelligible fashion and written in standard English?

Reviewer #1: Yes

Reviewer #2: Yes

5. Review Comments to the Author

Reviewer #1: This is a follow-up study of RF induced heating of DBS leads. The authors have previously published differences in RF induced heating of DBS leads between a horizontal, closed-bore MRI system and a vertical, open-bore system in two MRM publications (Ref 11 and 12). The current work extends the previous studies with further fiber-optic measurements of temperature rise around a commercial DBS device implanted in an anthropomorphic phantom following 30 patient-derived configurations. In addition, the electric length of the lead was varied to explore transferability of the results to leads from other manufacturers.

The study is methodologically sound and of principle interest for the scientific and medical community.

However, I do have a few comments for the authors:

Introduction, page 4: The authors state that in the previous simulation studies “simplified DBS lead models” were used, and also the term “proof-of-concept studies” implicates that the previous work may be less comprehensive than the current one. However, when I looked at both MRM publications, I actually found the opposite. Especially in Ref 12 a total of 90 lead models were assessed compared to 30 in this work, and the authors already performed experimental measurements of temperature rise for 4 trajectories and three configurations. Maybe you could rephrase that part of the introduction, and also clearly describe whether the 30 assessed configurations here are a subset of the previous 90 configurations, or a totally new set that has not been explored before? If so, why was a new set needed? What is the difference between the old and new set? Since temperature measurements have been performed before, the first question for this paper has already been answered with the previous 12 test cases compared to an additional set of 30 in this paper. What is the added value?

Introduction & Methods: The authors compare simulation results of SAR with measured temperature rise. I think thermal simulations instead of SAR would provide much more insight and provide a better metric for comparison. Please comment.

Methods part 2.2, page 7: Why have you studied 27 trajectories that were contralateral to the IPG, but only 3 that were ipsilateral to the IPG? Three is a very, very low number, and this choice of different trajectories has not been explained. Even in terms of statistical evaluations a set of 5 cases is the absolute minimum. Please comment.

Methods part 2.2, page 8: The temperature rise was measured for less than 4 minutes only! Why was the measurement time so short? When I look at the curves I do not so a flattening of the curves that would justify an early abortion of the measurement, but a continuous increase in temperature. Please comment.

Figure references on page 9 and 11: Figures 4 and 5 are mixed up. Please revise their order.

Methods page 9, Table 3: If I understand correctly, the authors addressed their second aim of the study, i.e. the transferability of the previous simulation results to leads from other manufacturers by three configurations only. A straight wire was compared to helical wires with pitches of 1 mm and 2 mm, but not for 30 different trajectories. Correct? Is that really enough evidence to generalize your results to DBS devices from other manufacturers? Please do not get me wrong. I understand that all the simulations are time-consuming and extensive, but I do see a mismatch in answering your two questions here. First, I see that already 90 lead trajectories have been compared between both scanner systems along with 12 selected temperature measurements in the 2021 MRM publication, and still, you add another 30 measurements in this study. On the other hand, for the second question 3 simulations without supporting experimental measurements should be sufficient.

Reviewer #2: This well-written manuscript investigates the possibility of safe MRI scanning the patients with DBS lead in vertical, open-bore scanners. The first objective of the authors is to explore the validity of their previously-present numerical results using a vertical scanner and a commercial DBS lead. Their second objective is to extend the outcomes of this work to different types of commercial DBS leads from other vendors. Their experimental results show more than ten folds reduction in maximum temperature rise if a vertical scanner is used instead of a horizontal scanner. Although I believe this work contains critical scientific and practical information that may significantly impact the pathway toward DBS patients’ imaging, two major concerns and several minor issues need to be addressed before publication.

Major:

1. One of the main objectives of this manuscript is to extend the presented experimental results to other commercial DBS leads. To accomplish this objective, numerical simulations of a DBS lead with different internal wiring structures were exploited. However, only one trajectory (not to mention that the trajectory generates a large difference in Tmax between RF exposures in the horizontal versus vertical scanners) was used for this purpose. To be inclusive, please perform a similar analysis for all trajectories (30 available trajectories). Presenting a graphic similar to Fig. 4C (30 trajectories, each containing 6 bars: 2 Scanners × 3 Wiring Structures) would be ideal.

2. In heating studies (TSE sequence), the scanner-reported B1rms value is chosen as the control metric and set to a fixed value for all cases (4uT). To the best of the reviewer’s knowledge (please explain if it is otherwise), B1rms is a pre-determined value by the scanner’s vendor based on numerical simulations with the corresponding coil model. If this is the case, the reported value does not represent a fair comparison criterion for different scanners from different vendors. At this point, I suggest following either of the pathways below:

a. Take the DBS lead out of the phantom � Perform B1 or Flip Angle mapping on the central transverse slice in both scanners with the “B1rms=4uT” setting � Compare the obtained B1 maps

b. Use the total input power or global SAR reported by the scanner as the control metric instead of the B1rms

Minor:

3. Introduction (P3.Para3): Please include a brief description of RMS B1+

4. Introduction (P4.Para2): “… which produces a notably different field distribution …” please change the “field” to either “EM field” or “electric field”. Please also add a proper reference for this statement (e.g., your previous publication: doi.org/10.1002/mrm.28818).

5. Materials and Methods (P5.Para4): Is the dataset of segmented models (30 realistic trajectories) publicly available?

6. Materials and Methods (P6.Para2): Please also add the relative permittivity of the remaining head-torso phantom.

7. Materials and Methods (P6.Para3): “… heterogenous human body model …”. Is it the DUKE model? Please add a proper reference.

8. Materials and Methods (P6.Para3): “… Supplementary Figures S3 and S4…”. Although it is presented as a good agreement, some noticeable discrepancies are visible. Since the hot spot ring is fully closed in the phantom, my first guess is a higher permittivity in the phantom (shorter wavelength). Please comment on this.

9. Materials and Methods (P7.Para2): I understand that two temperature probes were attached to the lead. However, only one temperature rise is given for each case in Fig. 4A. What is the use of the other probe?

10. Materials and Methods (P7.Para2): Is the IPG set to the MR-mode as instructed by the vendor (Abbott)? If not, please investigate at least one case in MR-mode (both scanners).

11. Materials and Methods (P9.Para2): Fig. 4A and 4B are mislabeled � Fig. 5A and 5B

12. Materials and Methods (P9.Para2): “… mean B1+ of 4uT on a central transverse …”. Is that adjusted to be 4uT in the presence or absence of the lead?

13. Materials and Methods (P9.Para3); Equation 1: Please replace x with bar{r}. It can easily be misinterpreted as the x-coordinate of the cartesian coordinates system.

14. Materials and Methods (P10.Para1); Equation 2: There are some issues/errors in this definition as well as its utility:

a. The result of the given integration has a time dependence. However, peak-to-peak voltage is usually defined as a constant value between the min and max of a monochromatic signal.

b. Since the voltage is defined as the difference in the electric potential between two points, the corresponding points need to be explicitly mentioned. If presumably, the integration is along the lead, then the defined voltage is between two ends of the DBS lead. However, the quasi-static assumption does not hold for such an elongated lead, and the “induced EMF” loses its meaning. Since this voltage is presented as a metric to compare the RF heating in different cases, please either justify it or change the metric (e.g., induced current, pSAR, etc).

15. Results (P11.Para1): Fig 4 � Please correct the order of Fig. 4 and 5

16. Fig 5: Is the shield of the birdcage coil included in simulations? Is the vertical coil shielded?

17. Results (P11.Para2): Fig 5:

a. Is the vector presentation corresponding to the incident E-field? Please mention it.

b. The simulation software computes the total E-field (incident + scattered). How do you distinguish the incident field?

c. Is the color presentation over the lead corresponding to the tangential E-field? Please mention it.

18. Discussion (P12.Para3): “… conventional solenoidal birdcage coils …” Birdcage coils are not solenoidal structures. Please remove the term.

19. Discussion: Although it is an interesting idea to use a helical shape to mimic the internal wiring structure of DBS leads, it is not accurate modeling, especially for multichannel leads (with complex wiring strategies). Please discuss this matter.

6. PLOS authors have the option to publish the peer review history of their article (what does this mean?). If published, this will include your full peer review and any attached files.

Reviewer #1: No

Reviewer #2: No

---

## [Author Response · Author response to Decision Letter 0]

12 Oct 2022

A separate document has been uploaded to respond to reviewer comments

---

## [Decision Letter · Decision Letter 1]

14 Nov 2022

A comparative study of RF heating of deep brain stimulation devices in vertical vs. horizontal MRI systems

PONE-D-22-22994R1

Dear Dr. Golestanirad,

We’re pleased to inform you that your manuscript has been judged scientifically suitable for publication and will be formally accepted for publication once it meets all outstanding technical requirements.

Kind regards,

Stephan Orzada

Academic Editor

PLOS ONE

Additional Editor Comments (optional):

Thank you for your careful revision. The reviewers both thank you for your affords. There is only a minor remark on one figure.

Reviewers' comments:

Reviewer's Responses to Questions

**Comments to the Author**

1. If the authors have adequately addressed your comments raised in a previous round of review and you feel that this manuscript is now acceptable for publication, you may indicate that here to bypass the “Comments to the Author” section, enter your conflict of interest statement in the “Confidential to Editor” section, and submit your "Accept" recommendation.

Reviewer #1: All comments have been addressed

Reviewer #2: (No Response)

2. Is the manuscript technically sound, and do the data support the conclusions?

Reviewer #1: Yes

Reviewer #2: Yes

3. Has the statistical analysis been performed appropriately and rigorously? 

Reviewer #1: Yes

Reviewer #2: Yes

4. Have the authors made all data underlying the findings in their manuscript fully available?

Reviewer #1: Yes

Reviewer #2: Yes

5. Is the manuscript presented in an intelligible fashion and written in standard English?

Reviewer #1: Yes

Reviewer #2: Yes

6. Review Comments to the Author

Reviewer #1: Thank you for your very detailed answers and clarifications. The manuscript has improved from the additional experiments and simulations. Congratulations for your work.

Reviewer #2: I would like to thank the authors for their careful revision, all my questions have been answered satisfactorily. Upon reading the revised document, I have only one remaining minor comment:

Fig. 4a: The rectangular box is labeled as a “shield”. Since this is not the case in the real system, I believe the box represents the region/boundary in the simulation environment. Please include a realistic cylindrical shield with the corresponding label.

7. PLOS authors have the option to publish the peer review history of their article (what does this mean?). If published, this will include your full peer review and any attached files.

Reviewer #1: No

Reviewer #2: No

---

## [Editor Report · Acceptance letter]

1 Dec 2022

PONE-D-22-22994R1 

A comparative study of RF heating of deep brain stimulation devices in vertical vs. horizontal MRI systems 

Dear Dr. Golestanirad:

I'm pleased to inform you that your manuscript has been deemed suitable for publication in PLOS ONE. Congratulations! Your manuscript is now with our production department. 

Kind regards, 

on behalf of

Dr. Stephan Orzada 

Academic Editor

PLOS ONE